# TOOLSCAN: A BENCHMARK FOR CHARACTERIZING ERRORS IN TOOL-USE LLMS

**Shirley Kokane**[†]**, Ming Zhu**[†]**, Tulika Awalgaonkar**[†]**, Jianguo Zhang**[†]**, Akshara Prabhakar**[†]**,
Thai Hoang**[†]**, Zuxin Liu**[†]**, Rithesh Murthy**[†]**, Liangwei Yang**[†]**, Weiran Yao**[†]**, Juntao Tan**[†]**,
Zhiwei Liu**[†]**, Juan Carlos Niebles**[†]**, Huan Wang**[†]**,
Shelby Heinecke**[†]**, Caiming Xiong**[†]**, Silvio Savarese**[†]

[†]Salesforce AI Research

## ABSTRACT

Evaluating Large Language Models (LLMs) is one of the most critical aspects of building a performant compound AI system. Since the output from LLMs propagate to downstream steps, identifying LLM errors is crucial to system performance. A common task for LLMs in AI systems is tool use. While there are several benchmark environments for evaluating LLMs on this task, they typically only give a success rate without any explanation of the failure cases. To solve this problem, we introduce TOOLSCAN, a new benchmark to identify error patterns in LLM output on tool-use tasks. Our benchmark data set comprises of queries from diverse environments that can be used to test for the presence of seven newly characterized error patterns. Using TOOLSCAN, we show that even the most prominent LLMs exhibit these error patterns in their outputs. Researchers can use these insights from TOOLSCAN to guide their error mitigation strategies.

## 1 INTRODUCTION

An emerging use case for LLMs in AI systems is tool use. When LLMs are equipped with context and a list of tools, such as APIs, databases, and applications, LLMs can generate a sequence of function calls to solve a given task. Using LLMs to perform tasks via APIs involves more complex reasoning and instruction following abilities of LLMs, and there are several emerging methods that aim to improve those abilities in order to help LLMs adapt to new tasks and situations. These methods (Zhang et al., 2024; Li et al., 2023) help enhance the model's understanding of user preferences, along with its skills in logical reasoning, with the goal of increasing their overall effectiveness. The basic technique involves using guiding prompts to provide LLMs with instructions and information about the context, enabling them to generate actions to solve complex tasks. Besides, some techniques (Rafailov et al., 2024; Jing et al., 2019) focus on dedicated training methods to turn LLMs into highly capable agents.

Significant progress has been made in evaluating LLMs on tool use tasks, as demonstrated by several benchmarks such as TOOLBENCH (Guo et al., 2024), AGENTBOARD (Ma et al., 2024), MINT-BENCH (Wang et al., 2023), and AgentLite (Liu et al., 2024). Collectively, the currently available benchmarks span tool use scenarios in a broad range of environments such as weather (Open-Meteo) and movie (The-Movie-DB). However, most evaluation benchmarks typically only calculate the success rate, measuring how often the final output of the LLMs aligns with the expected output. The AGENTBOARD benchmark includes a success rate metric that specifically evaluates the model's final output against the expected ground truth. Moreover, the tool-related dataset within this benchmark is limited in scope. Conversely, the TOOLBENCH benchmark features a more extensive dataset, encompassing a wide range of tasks across different instructions, tools, and categories. Nevertheless, TOOLBENCH is similarly restricted to assessing whether the final outcome has been achieved. Although leading LLMs demonstrate similar overall performance as reported in (Guo et al., 2024), a closer analysis reveals distinct underlying errors in their behavior. To drive continuous improvements in LLM performance, it is essential to thoroughly understand these failure cases.

Table 1: Comprehensive Comparison of TOOLSCAN across critical evaluation requirements. TOOLSCAN supersedes on all the provided principles, focusing on realistic multi-turn interactions.

| Points | ToolBench | AgentBoard | MintBench | Gorilla | ToolScan |
|---|---|---|---|---|---|
| Varied Unique APIs | ✓ | ✓ | ✗ | ✗ | ✓ |
| Query Diversity | ✓ | ✗ | ✗ | ✓ | ✓ |
| Error Pattern Analysis | ✗ | ✗ | ✗ | ✓ | ✓ |
| Feedback Mechanism | ✗ | ✓ | ✓ | ✗ | ✓ |
| Multiple Ground Truth Trajectories | ✗ | ✗ | ✗ | ✗ | ✓ |
| Number of Tool-Use Queries | 200 | 60 | 134 | 200 | 150 |
| Number of Tasks | 105 | 3 | 1 | 9 | 30 |

In an effort to address the limitations of the existing benchmarks, we present TOOLSCAN, a benchmark created to characterize common errors in tool-use LLMs and provide detailed diagnostic feedback to help improve them. TOOLSCAN comprises of 10 distinct environment categories, including Tools, Movies, Travel, Sports, Entertainment, Data, Social, Media, Weather and Video Images, making it one of the most comprehensive evaluation environments. It also incorporates over 30 different tasks specifically aimed at tool agents, such as entertainment, sports, and weather tasks.

During this study, we delineate seven commonly occurring error patterns displayed by LLMs when engaged in tool-use tasks. In order to bring consistency to the evaluation of these error patterns across varying LLMs, we build a comprehensive evaluation framework. The framework comprises a comprehensive error pattern analysis and feedback mechanism tailored for various agents operating in diverse environments, all within a unified format. The feedback mechanism enables agents to refine their actions based on fundamental tool-calling criteria. This setup offers valuable insights into the different error patterns encountered by the models.

Furthermore, the TOOLSCAN evaluation dataset comprised of 150 queries can be employed to identify these error patterns in the output of LLMs during tool-use tasks. These queries have been annotated by humans to highlight multiple pathways to reach a given objective. Subsequently, we showcase the capacity of TOOLSCAN to identify error patterns in various leading LLMs.

The contributions of this paper can be summarized as follows:

- We introduce TOOLSCAN, a comprehensive benchmark covering 10 environment categories and over 30 tasks, specifically designed to provide detailed diagnostic feedback on tool-use tasks in LLMs.

- We identify seven common error patterns in LLM tool-use tasks and create an evaluation framework that analyzes these errors consistently across different agents and environments.

- We provide a 150-query human-annotated dataset to detect error patterns and demonstrate TOOLSCAN's effectiveness through a case study involving several leading LLMs. Number of tool-use queries in our dataset are similar to other datasets while providing better coverage on diversity, feedback mechanism etc.

## 2 RELATED WORK

The current benchmarks exhibit notable strengths and weaknesses in evaluating model performance as a tool-use agent. Benchmarks like GORILLA (Patil et al., 2023), AGENTBOARD (Ma et al., 2024), TOOLBENCH (Guo et al., 2024), TOOLEYES (Ye et al., 2024), and MINTBENCH (Wang et al., 2023) are remarkably utilized in examining model interaction ability, reasoning and planning ability. While some benchmarks like TOOLBENCH, AGENTBOARD, and MINTBENCH are effective in managing multi-turn interactions, others such as GORILLA excel in handling diverse queries, especially when dealing with constraints and irrelevant information. GORILLA also offers some error pattern analysis and are well-suited for complex queries that require a wide range of API capabilities but has limited diverse tasks.

However, these evaluations often fail to consider the model's existing knowledge, leading to unnecessary and repetitive API calls without specific prompt instructions. Furthermore, the benchmarks like AGENTBOARD place undue emphasis on the sequence of sub-goals, sometimes misjudging a

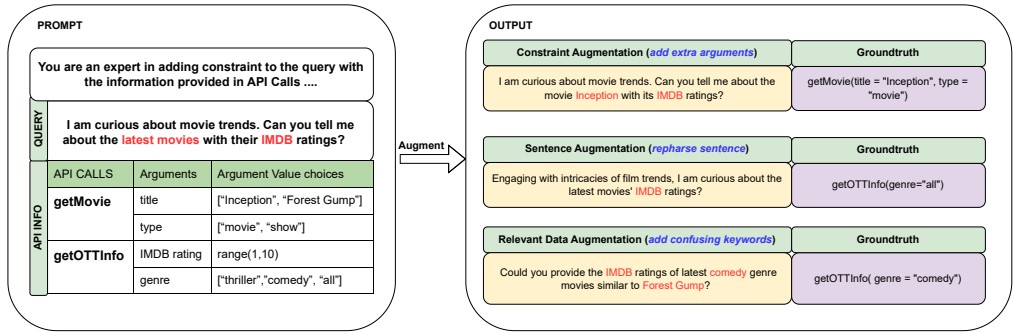

Figure 1: Query Generation Workflow for TOOLSCAN. Query and API Info collected from Open Source Toolsets are given as input to an LLM with a system prompt. The LLM is asked to generate augmented queries along with the API Calls required to solve that query.

model's progress when redundant sub-goals are involved. This rigid focus can result in an inaccurate assessment of the model's true capabilities in task completion.

We address all these current issues in the existing benchmark, where our focus is more on assessing the quality of the action generated by the model given the constraints in the query. Our emphasis is on understanding some critical errors of the model such as hallucinations in actions w.r.t API names or argument names, performing redundant repetitive calls, invalid format issues etc. It is important to quantify such issues to get a deeper insight into the improvements required in the model.

## 3 ERROR PATTERNS IN TOOL-USE

It's widely understood that LLMs are prone to output errors such as hallucinations, inconsistencies, errors, etc (Wei et al., 2022; 2024; Lu et al., 2024). What is less studied are the errors exhibited in tool-use scenarios. In tool-use scenarios, typically the LLM outputs one or more function calls, including necessary arguments for the execution of the functions. For example, an LLM output for using a web search API to learn about the latest fashion can be as the following: `Search[query="latest fashion", top_k=10]`, where `Search` indicates the tool and `query` and `top_k` are the context-specific arguments for executing the tool call.

Note that these tool calls are particularly brittle - even a slight change to an argument, a missing argument, or incorrect tool call can produce drastically different and incorrect results downstream. Given the importance of catching these tool call errors, in this work, we characterize systematic errors generated by LLMs on tool-use tasks into seven critical error types.

- **Insufficient API Calls** (IAC): Unable to generate sufficient API calls, hence unable to completely fulfill the tasks provided in a query.

- **Incorrect Argument Value** (IAV): Generates incorrect argument values. This also includes exclusion of required arguments.

- **Incorrect Argument Name** (IAN): Hallucinates argument names.

- **Incorrect Argument Type** (IAT): Generates incorrect argument type.

- **Repeated API Calls** (RAC): Generates the exact same API Call repeatedly, causing redundant calls.

- **Incorrect Function Name** (IFN) : Hallucinates function names, that are not part of the API list.

- **Invalid Format Error** (IFE) : Generates inappropriate format.

Table 2: Examples of Error Patterns Identified in Models. GT stands for Ground Truth Labels, MO stands for Model Output.

| Query | Error | Model | Output |
|---|---|---|---|
| I need jokes sorted by score in ascending order of the category food? Can you provide me with that? | IAV | GPT-3.5 | **GT**: *getJokeCategory(category='food', sortby= 'asc')* |
| | | | **MO**: *getJokeCategory(category='workplace', sortby = 'asc')* |
| I want the genre, runtime, IMDb rating, language for 'Endgame'? Also, include the streaming platforms available in the US. | IAC | GPT-4 | **GT**: *getMovie(title='Endgame',type='movie'), getMovieProvider(region = 'US')* |
| | | | **MO**: *getMovie(title='Endgame',type='movie')* |
| I need the titles and release years for horror movies with a minimum IMDb of 7 released between 1970 - 2020. | IAN | MLlama | **GT**: *getMovieAdv(genre = 'horror', startyear = 1970, endyear = 2020, minimdb = 7)* |
| | | | **MO**: *getMovieAdv(genre='horror', releasestart=1970 , releaseend=2020, imdbrating =7)* |
| I need to check if the domain 'business.com' is available or not | IAT | xLAM | **GT**: *checkDomainAvailability(domain = 'business.com', availableonly=True)* |
| | | | **MO**: *checkDomainAvailability(domain= 'business.com' , availableonly="True")* |

## 4 TOOLSCAN DATASET

The brittle nature of LLM-generated tool calls highlights the need to quickly discover errors in LLM-generated outputs. To quickly discover tool-calling errors in LLMs, we propose a new dataset, TOOLSCAN. It is created through a series of effective steps aimed at generating complex queries.

The process begins by gathering seed queries for each API environment, sourced from benchmark tests conducted on Toolsets like TOOLBENCH(Guo et al., 2024) and AGENTBOARD(Ma et al., 2024). These initial seed queries are then augmented using the following methods also shown in Figure 1.

**Constraint based Query Generation:** Utilizing prompt engineering techniques, we incorporate detailed descriptions of each argument and its corresponding argument value choices to enhance the original query using GPT-4. We introduce method by adding multiple arguments and their options, making the query even more complex.

**Sentence Transformation based Query Generation:** On seed queries, we instruct GPT-4 to modify sentences while maintaining the same context. This means keeping the contextual requirements unchanged while altering the sentences.

**Relevant Data based Query Generation:** On seed queries, we instruct GPT-4 to introduce unnecessary information around the keywords while maintaining the original context of the query.

Based on our required criteria of creating complicated queries we generate 600 queries. We filter these queries based on feasibility evaluation, constraint validation and diversity. For example we remove infeasible queries such as incorrect function name, incorrect arguments, etc which leaves about 150 queries. More details in Appendix A.1.4.

Table 3 describes the distribution of queries across different environment APIs. We also list down the average number of interactions along with the queries sampled per environment after augmentation.

| Environments | Sampled Queries | Avg. Turns | Avg. APIs |
|---|---|---|---|
| Patent | 20 | 6 | 7.2 |
| Movies | 30 | 11 | 11.5 |
| Travel | 5 | 8 | 5.6 |
| Sports | 8 | 4 | 4.4 |
| Tools | 12 | 5 | 4.8 |
| Data | 14 | 7 | 4.0 |
| Social | 16 | 7 | 3.0 |
| Media | 11 | 6 | 5.5 |
| Spaceflight | 25 | 9 | 10.0 |
| Video Images | 9 | 5 | 6.0 |
| ToolScan | 150 | 6.8 | 6.2 |

Table 3: Comparison of sampled queries, interactions, and APIs across environments.

## 5 FRAMEWORK

In the evaluation benchmark we propose for tool use, we ensure that the environments are deterministic. This allows the trajectory of the agents to depend solely on the policy and actions selected by the language model. Agents are given a description of the available tools and task instructions, including format requirements. The actions generated by the agents are reviewed by an inbuilt feedback mechanism designed to detect prominent errors. These include formatting issues, incorrect function names, incorrect argument names, and incorrect argument types. If no such errors are found, the action is executed within the corresponding environment and the feedback it generates is collected. This feedback offers insights into both the changes in state and any potential action errors.

As described in the AGENTBOARD paper (Ma et al., 2024), tool use interaction with an environment can be conceptualized as a finite-horizon Partially Observable Markov Decision Process (POMDP) (Bellman, 1957), starting with a known initial state, described by the tuple $< G, S, A, T, O >$, where $G$ is the goal, $S$ represents the set of possible states, $A$ comprises the available actions, $T$ is the transition model $T : S \times A \rightarrow S$, and $O$ includes the observation space (along with feedback from the environment). An agent, guided by policy $\pi$, makes decisions at every iteration x based on the goal g and a memory sequence $m_t = o_x, a_x, o_{x+1}, a_{x+1}, \ldots o_t$, where $0 \leq x < t$, which records the sequence of past actions and observations. The resulting agent's path, $\tau = [s_0, a_0, s_1, a_1, \ldots s_t]$, emerges from the policy and the environment's state transitions.

As shown in Figure 2, our benchmark follows a deterministic framework where the agent will begin with Instruction $i$, Query $q$, and a sequence of API-list $A$ (resembling to the action space), provided to successfully solve the provided query. The agent then based on the policy of the language model will perform an action $a_t$ in the current $s_t$, leading to a determined state transition $s_{t+1}$.

A constructive feedback mechanism $f$ is also incorporated to help assist the agent for the following issues:

- Verifies the generated action is correctly parsed and adheres to the format instructions provided in the prompt.

- Determines if the generated action is in the specified action space. If not, enumerates the list of available actions that can be used to resolve the query.

- Assesses whether the generated action is invoked with the appropriate arguments. If not, provide a detailed list of the available arguments for the chosen action, along with their respective descriptions.

- Ensures that the generated action is called with arguments of the correct type. If not, specify the correct argument type required for proper execution of the function.

Upon determining that the feedback is positive and no errors are detected in the action, the action is executed within the environment $e$ to obtain a new observation, denoted as $o_t + 1$. This observation is then fed back into the model to facilitate the subsequent set of appropriate actions.

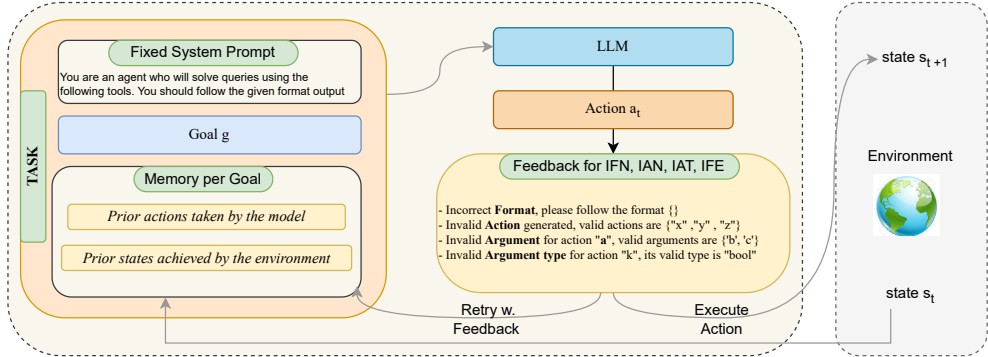

Figure 2: Systemic workflow of an agent interacting with the environment assessing the prior information and the predefined goal to determine its next action.

## 6   METRICS

Let $Q$ be the set of queries. For each query $q_i \in Q$, let $N$ represents the maximum steps the model is permitted to process the query, while $G_i$ denotes the Ground Truth labels for the $i^{th}$ query $q_i \in Q$. Metric definitions based on above are as follows:

- For error patterns IFN, IAN, IAT, IFE and RAC we calculate the percentage of API Calls executed without respective error as $\sum_{i=1}^{|Q|} 1 - \frac{e^i}{N}$, where $e^i$ is the number of API calls with the error pattern in $i^{th}$ query $q_i$.

- For error patterns IAC and IAV, let $\{g \in G_i\}$ be the set of possible trajectories for query $q_i$. Then we define our metric as for each error pattern as

$$\sum_{i=1}^{|Q|} 1 - \frac{\min_{g \in G_i} e_g^i}{N}$$

where $e_g^i$ are the number of API calls with error pattern when evaluated in comparison to ground truth trajectory $g \in G_i$ for query $q_i \in Q$.

- For each query, we calculate success rate by checking if the model is able to successfully complete the query.

It is crucial to emphasize that while the metrics indicate error patterns, their calculations are designed to be directly **proportional to the success rate**. Higher scores on the error metrics represent fewer issues generated by a model in that respective error pattern.

## 7   EXPERIMENTS

We've conducted a comprehensive assessment of several performant LLMs, including GPT-4 (Achiam et al., 2023), GPT-3.5-Turbo from OpenAI, Meta-Llama3-8b (Dubey et al., 2024), DeepSeek-R1-Distill Models (Guo et al., 2025), Code-Llama-13B (Rozière et al., 2024), Vicuna-13B-16k (Chiang et al., 2023), Mistral and Mixtral models from Mistral AI (Jiang et al., 2023) (Jiang et al., 2024), and xLAM (Zhang et al., 2024). Interestingly, while these models all perform well on TOOLBENCH(Guo et al., 2024), their failure patterns are different. Details of the error patterns discovered using TOOLSCAN are shown in Table 2.

We incorporate a one-shot in-context example along with task-specific instructions in our prompt setup. For each open-weight model, we evaluate the chat-optimized version when available, unless otherwise indicated. For models without a dedicated chat version, we apply standardized prompt templates tailored for publicly accessible versions. We ensure that each prompt includes precise format instructions to enable the model to generate API calls in a structured and parsable format. To account for server inconsistencies, we execute multiple rounds per API call, thus minimizing potential disruptions and ensuring consistent evaluation outcomes.

## 8   RESULTS

From our results in Table 4, we observe that LLMs specifically fine-tuned for API calling are generally less likely to produce false information, often known as "hallucination" when dealing with function names and argument names. On the contrary, models like Vicuna (Chiang et al., 2023) (Zheng et al., 2024) that are designed for chat-based interactions have a higher rate of inaccuracies, particularly incorrect argument names and function names. It suggests that the tuning process, specifically for API calls, can lead to a substantial reduction in hallucination.

We observe that Insufficient API Calls (IAC) is the most common error pattern. Most models have a significantly low IAC score. We observed two reasons for it: (1) Models like GPT only solves part of the query. If there are multiple questions asked in the same query, GPT ends after answering the fist query. (2) Models are unable to distinguish similar APIs and ignore the given constraints. For example, model will pick `get_patent_with_title` instead of `get_patent_with_title_and_date` and hence only return the patents with title and not date.

| Model | Success Rate | IAC | RAC | IAV | IFE | IAT | IAN | IFN |
|---|---|---|---|---|---|---|---|---|
| GPT-4-0125-preview | **0.73** | **0.84** | **1.00** | **0.94** | **1.00** | 0.93 | **0.94** | **0.94** |
| xLAM-8x7B | 0.72 | 0.79 | 0.95 | 0.92 | **1.00** | **0.95** | 0.93 | 0.91 |
| GPT-4o-turbo-2024-05-13 | 0.7 | 0.59 | 0.93 | 0.88 | 0.94 | 0.92 | 0.88 | 0.92 |
| GPT-3.5-turbo-1106 | 0.67 | 0.7 | 0.94 | 0.86 | 0.95 | 0.94 | 0.93 | 0.91 |
| xLAM-7B | 0.64 | 0.78 | 0.98 | 0.93 | 0.98 | **0.95** | 0.86 | 0.91 |
| Code-Llama-13B | 0.41 | 0.43 | 0.69 | 0.7 | 0.66 | 0.72 | 0.7 | 0.68 |
| Mixtral-8x22B-Instruct-v0.1 | 0.4 | 0.7 | 0.92 | 0.62 | 0.99 | 0.6 | 0.81 | 0.78 |
| DeepSeek-R1-Distill-Qwen-14B | 0.32 | 0.47 | 0.66 | 0.63 | 0.43 | 0.73 | 0.72 | 0.73 |
| Meta-Llama3-8b | 0.27 | 0.58 | 0.4 | 0.42 | 0.71 | 0.49 | 0.42 | 0.64 |
| Mistral-7B-Instruct-v0.1 | 0.24 | 0.49 | 0.54 | 0.21 | 0.23 | 0.52 | 0.47 | 0.6 |
| Vicuna-13B-16k | 0.16 | 0.27 | 0.26 | 0.61 | 0.69 | 0.39 | 0.4 | 0.5 |
| DeepSeek-R1-Distill-Qwen-7B | 0.11 | 0.19 | 0.31 | 0.35 | 0.11 | 0.23 | 0.37 | 0.37 |
| Mixtral-8x7B-Instruct-v0.1 | 0.1 | 0.11 | 0.12 | 0.11 | 0.1 | 0.15 | 0.12 | 0.2 |

Table 4: Results of ToolScan Benchmark across several performant LLMs on the proposed error metrics. *IAC, IAV, RAC, IFE, IAT, IAN, IFN* are error metrics occurring in each API Calls. We show the success rate as well as the percentage of API calls not having the error metrics (Higher is Better).

We also observe that smaller models which are not tuned for function calling such as Meta-Llama3-8b, DeepSeek-R1-Distill-Qwen-7B etc. perform Repeated API Calls (RAC) despite clear instructions to avoid repetition. They get stuck in a loop based on their internal belief of solving the query and call the same API multiple times even if they are failing. We also see a co-relation between RAC and IAC between these models which suggests that RAC often leads to IAC.

For Incorrect Argument Value (IAV), we observe that models tend to ignore optional arguments like sorting, limit etc and tends to go with the default values set in the functions. This can be seen in models like Code-Llama-13B, Mixtral-8x22B-Instruct-v0.1 and DeepSeek-R1-Distill-Qwen-14B. This error is especially prevalent in smaller models like Mistral 7B because it ignores "required" arguments explicitly mentioned in the APIList.

Models like Mixtral-8x22B-Instruct-v0.1 also face errors in specifying correct argument type, confusing string for int and vice-versa. This increase the Incorrect Argument Type (IAT) error and hence reducing IAT score. Depending on the type of training done on models they tend to exhibit different outcomes in error patterns. For instance, DeepSeek-R1 distill models are specifically trained on MATH tasks with profound focus on improving their reasoning. Hence, on tool-use tasks these models face difficulty in following format instructions (low IFE) and spend significant chunk of tokens in the thinking process.

Most striking is the superior performance of GPT-4, achieving the highest success rate among all models evaluated. This could potentially suggest that larger models harnessing the power of the scaling law and quality pre-training in the base model remain critical to the success of agent applications. Another noteworthy observation is the exemplary performance of the Code-Llama-13B model (Rozière et al., 2024), surpassing many other generically-purposed models. Our hypothesis is that there may be some inherent similarity between coding-associated tasks and function calls. And that could be beneficial for models designed for coding purposes, thereby enhancing their performance on function-calling tasks.

## 8.1 ABLATION STUDY

### 8.1.1 IMPACT OF IRRELEVANCE ON MODEL PERFORMANCE

We did an in-depth examination on how irrelevant constraints affect model accuracy in generating API calls across two distinct environments. Irrelevance is defined as additional, unrelated requirements conflicting with the given API environment. Our analysis measures the impact on success rate and the tendency to hallucinate fictitious API calls.

Results indicate that most models produce more errors in Incorrect Function Name (IFN), posing deployment challenges in the real-world scenario. This underscores the need for improved handling of unexpected inputs. Figure 3 (Left) highlights that the percentage of queries without IFN decreases when irrelevant constraints are introduced in the query. This underscores the importance of our benchmark in uncovering the exact reason for the failure of the models.

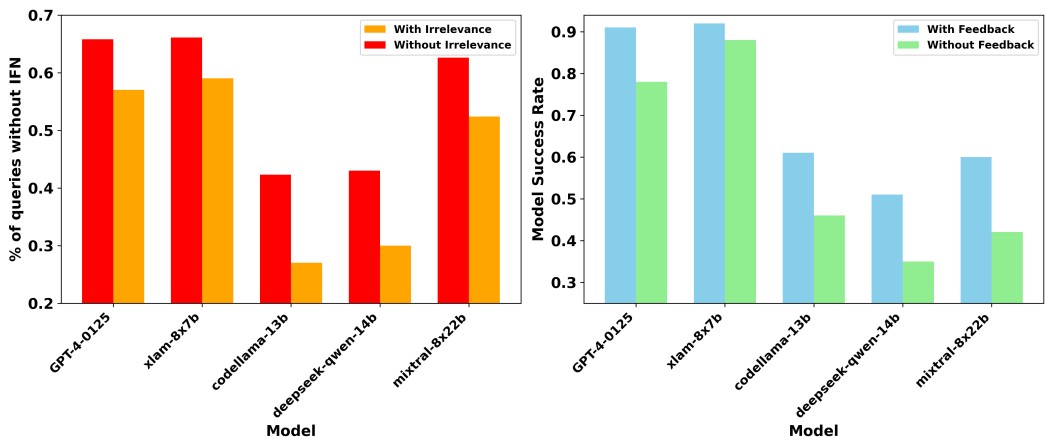

Figure 3: Left: We see that percent of queries with Incorrect Function Name (IFN) error is higher when we augment queries with irrelevant terms (Higher is Better). Right: We see that model success rate is higher when model is provided with feedback which helps it to correct itself (Higher is Better).

### 8.1.2 IMPACT OF FEEDBACK ON MODEL PERFORMANCE

We also performed an ablation study to assess the importance of the constructive feedback mechanism introduced in the TOOLSCAN benchmark. As shown in Figure 3 (Right), the feedback mechanism plays a pivotal role in enabling the model to refine its actions by concentrating on essential aspects, such as correctly identifying API names, selecting appropriate API arguments, and adhering to the expected data types for those arguments.

In scenarios where the feedback mechanism was absent, the model frequently repeats errors, such as incorrect function names, leading to a significant reduction in its task completion effectiveness. Without guidance, the model struggled to recover from initial mistakes, which compounded its challenges in achieving desired outcomes. Conversely, with the feedback mechanism in place, the model exhibited immediate improvements in subsequent interactions. For example, it was able to adjust argument types or incorrect/mismatched argument names efficiently in response to errors. These results emphasize the critical role of feedback in enhancing the adaptability and accuracy of models, particularly in iterative and error-prone processes.

### 8.1.3 IMPACT OF OUTPUT FORMAT METHOD ON MODEL PERFORMANCE

We investigated the impact of structured versus non-structured output formats on the performance of LLMs. A related study by (Tam et al., 2024) examined open-ended benchmarks such as (Cobbe et al., 2021) and (Fansi Tchango et al., 2022), evaluating the effects of various output formats—including loose string, JSON, and YAML—on model performance. Their findings indicate that structured formats negatively impact an LLM's ability to generate and reason effectively. However, their study does not explicitly identify the underlying cause of this phenomenon. We hypothesize that this performance degradation is attributed to the increased token count required in structured

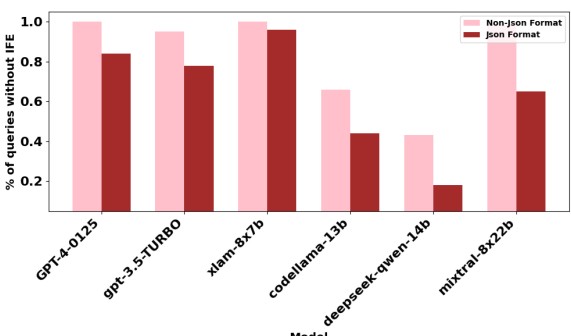

Figure 4: Comparison of Structured Unstructured Format versus Incorrect Format Error (Higher is Better).

formats, as additional spaces, colons, and indentation contribute to token inflation. This, in turn, reduces the proportion of meaningful tokens dedicated to the core task.

Our results in Figure 4 align with this observation, demonstrating a similar decline in performance due to Incorrect Format Errors (IFE) in our function-calling task. Notably, our task presents a higher

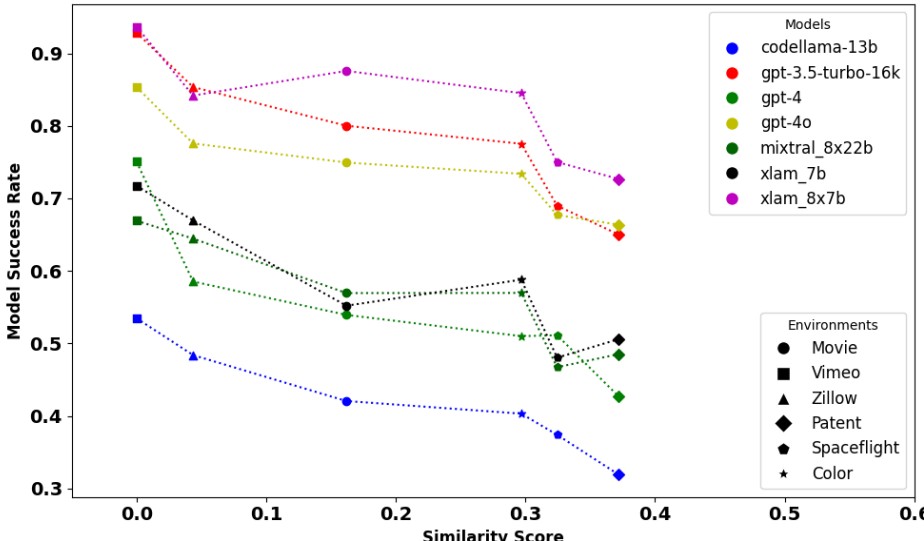

Figure 5: We see that environments which similar APIs tend to confuse the models leading to lower performance (Higher is Better).

level of complexity compared to the benchmarks in (Tam et al., 2024), as it involves a significantly larger API space with a greater variety of function calls, argument names, and argument types.

### 8.1.4   IMPACT OF API COMPLEXITY ON MODEL PERFORMANCE

The following study investigated how structural similarities between different APIs affect model decision-making and accuracy across various environments. To conduct this analysis, we isolated unique, descriptive tokens in each API function name by removing tokens shared across all names within an environment. This approach enabled us to compute a more accurate similarity measure using the Jaccard metric, which reflects how structurally similar API names are within each environment. This metric is advantageous as it assigns higher similarity scores to structurally similar APIs, which may differ only slightly, while scoring dissimilar, functionally distinct APIs lower. For details on how we remove shared tokens and compute the metric, refer to Appendix B.4.

The findings, shown in Figure 5, align with our hypothesis that environments with structurally similar API names are associated with lower model accuracy. Specifically, we observed a pronounced effect on the Insufficient API Calls metric, illustrating that models tend to misinterpret similar API names and, as a result, frequently select incorrect APIs. This study highlights the significance of breaking down error patterns to pinpoint the specific types of errors models make, ultimately improving transparency and guiding strategies to increase robustness in API-driven tasks.

## 9   CONCLUSION

This paper introduced TOOLSCAN, a benchmark for evaluating LLMs on tool-use tasks with a focus on error patterns and feedback. Our findings highlight the importance of task-specific fine-tuning, as models optimized for API calls perform more reliably and reduce hallucinations compared to chat-oriented models. Larger models, like GPT-4, demonstrate superior performance, affirming the significance of scaling and quality pre-training. Additionally, we showed that the feedback mechanism, relevance and unstructured output format in TOOLSCAN plays a crucial role in improving the model accuracy, guiding models to correct their function-calling behavior.

In this work, we recognize the importance of capturing the significant error patterns of the model, as addressing these failures in planning and function-calling can lead to long-term improvements in the model effectiveness, hence enhancing the reasoning capabilities of LLMs. By analyzing these metrics, developers can identify specific weaknesses in language models and focus on improvements in those key areas. This targeted approach can help enhance the overall performance and reliability

of the models. For future work we are going to design environments involving multiple family of actions to develop a more robust testing environment.

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

# A APPENDIX

## A.1 PROMPTS FOR QUERY AUGMENTATION

In the following section, we list all the prompts used in the process of query generation.

### A.1.1 Constraint Based Augmentation

Table 5 here describes the constraint based query generation prompt with a sample few-shot examples from the spaceflight environment. For every new environment we have its specific set of few-shot examples.

### A.1.2 Sentence Transform Based Augmentation

Table 6 here describes the sentence transform based query generation prompt.

### A.1.3 Irrelevance Based Augmentation

Table 7 here describes the irrelevance based query generation prompt.

### A.1.4 Query Data Verification

To generate high-quality queries from the proposed augmentation methods, we employ a systematic refinement process guided by multiple important criteria. Specifically, we generate the tool calls required to solve query by an LLM proficient in function-calling to evaluate and refine the generated queries. This process includes the following steps:

- **Feasibility Evaluation**: We ensure that all requisites of the generated queries are solvable using the available API list. This involves verifying the compatibility of the queries with the functionalities provided by the APIs.
- **Constraint Validation**: The quality of constraints is assessed by leveraging the APIs' execution capabilities. We check whether the arguments specified in the queries are feasible, executable, and capable of yielding significant outputs.
- **Manual Review**: To ensure diversity and relevance, we manually assess the difficulty level of each selected query. This step helps to curate a set of queries that are representative of various use cases and complexity levels.

By combining automated evaluation with manual review, we aim to produce a diverse set of high-quality queries that are both executable and meaningful in the context of the given API list.

## A.2 Environments Selection for ToolScan

We curate different functions with their respective parameter requirements for each environment using the following open-source APIs:

- Patent API
- Spaceflight API
- Movie API
- Weather API
- Other APIs

# B Method

## B.1 Query Selection for ToolScan

Based on the feasibility of solving the query given the provided APIs, we randomly select 150 base queries. We make sure that we don't introduce any duplicate or closely related queries generated by the augmentation.

### B.1.1 Queries Selected for Feedback Ablation Study

For the feedback vs no feedback ablation study we use the same 150 queries to analyse the impact of the specific feedback provided to the model for executing the query successfully. Within, the

average model score the major impact is caused on incorrect argument name, incorrect argument type and incorrect function name.

### B.1.2  QUERIES SELECTED FOR IRRELEVANCE ABLATION STUDY

For the irrelevance ablation study, we specifically gather queries from the irrelevance augmentation. We do this for the base queries from 3 environments (Spaceflight, Movie and Patent). In total, this ablation study is done using 60 queries.

### B.1.3  QUERIES SELECTED FOR OUTPUT FORMAT ABLATION STUDY

For the JSON vs no JSON ablation study we use the same 150 queries to analyse the impact of the structured format vs unstructured format method provided to the model for executing the query successfully. We have only shown the impact on incorrect format errors, but there is also a drop in other error metrics as well.

### B.1.4  QUERIES SELECTED FOR API COMPLEXITY ABLATION STUDY

For this study, we randomly collect 10 queries per environment to maintain unanimous averaging of scores across all the provided environments. The queries are selected from out original 150 base queries.

## B.2  PROMPT FOR BENCHMARK INFERENCE

Table 8 describes a sample prompt used during the inference in the Benchmark.

## B.3  SAMPLE TRAJECTORY OF THE BENCHMARK

Table 9 describes a sample trajectory and highlights over the specific feedback provided to execute the query.

## B.4  DETAILS ON API SIMILARITY IMPACT ON MODEL PERFORMANCE

For the API Similarity Calculation, lets take an example of Patent Enviornment with the following functions.

```
[get_patent_with_title,
get_patent_with_title_and_date,
get_patent_with_title_lte,
get_patent_with_title_neq,
get_patent_with_title_eq].
```

We remove the tokens common across the list of functions and convert the functions in the following way:

```
[patent_title,
patent_title_date,
patent_title_lte,
patent_title_neq,
patent_title_eq]
```

Post refinement we calculate the similarity of each function name w.r.t to another:

$$J(f_i, f_j) = \frac{|f_i \cap f_j|}{|f_i \cup f_j|} \tag{1}$$

for all unique pairs $(f_i, f_j)$, where $i < j$.

The overall average Jaccard similarity across all function name pairs is then given by:

$$J_{\text{avg}} = \frac{1}{\binom{n}{2}} \sum_{1 \leq i < j \leq n} J(f_i, f_j) \tag{2}$$

where $\binom{n}{2} = \frac{n(n-1)}{2}$ represents the total number of unique function name pairs.

Table 5: Prompts for Constraint Based Augmentation

You are an expert in augmenting agent's queries for given parameters. You will be given some few shot examples and a list of tools, with their descriptions and their required parameters and based on your knowledge you have to generate queries with constraints based on the tools provided.
Each of the answer is expected to follow the format below:
Query1: (constraint added query)
Query2: (constraint added query)
Queryn: ....

Your task is providing the queries with 3 or 4 constraints based on the list of tool calls available, using the following criteria:
1. Add additional constraints to the query but make sure the query has single fixed answer instead of a worded description.
2. Focus on making significant use of arguments of each tool call in an innovative way. Pay close attention on each Argument's Description and their input requirements while using them in the query.
3. Answer does not hallucinate and is relevant to the given tools list.
4. Innovate around using functions that are similar but have a very distinct difference.
5. Correctly follow the provided format.

FEWSHOT EXAMPLES:
**Query1**: Find articles from NASA or SpaceNews related to Artemis missions, not including those from European Spaceflight, limited to 2 results published between January 1st, 2021, and December 31st, 2021.
**Query2**: Retrieve the most recent article related to SpaceX from Spaceflight Now and the corresponding report with a summary mentioning "Falcon Heavy"; both published after January 1st, 2020.

Table 6: Prompts for Sentence Based Augmentation

You are an expert at refining an agent's queries by rephrasing sentences for clarity and precision without altering their intended meaning.

Each of the answer is expected to follow the format below:

Query1: (sentence transformed query)

Query2: (sentence transformed query)

Queryn: ....

You will receive some queries along with a list of tools, including their descriptions and required parameters. You will also be provided with an example of original vs augmented query. Your task is to transform each query's sentence structure, ensuring that the revised sentence preserves the same context and intent as the original. Follow these criteria to guide your rephrasing:

1. Maintain the query's original context and aim, ensuring that the rephrased sentence yields the same single, definitive answer as the original.

2. Keep the rephrased sentence relevant to the tools provided, without introducing any additional information or hallucination.

3. Pay close attention to each tool's specific parameters and descriptions, making sure the rephrased query aligns with the tools' intended use.

4. Follow the provided format precisely for each rephrased query.

ONE-SHOT EXAMPLE:

**Original Query**: Retrieve two articles focused on the Artemis missions, sourced only from NASA or SpaceNews, ensuring that no content originates from European Spaceflight, and limit the publication date to between January 1, 2021, and December 31, 2021.

**Augmented Query**: Find articles from NASA or SpaceNews related to Artemis missions, not including those from European Spaceflight, limited to 2 results published between January 1st, 2021, and December 31st, 2021.

Table 7: Prompts for Irrelevance Augmentation

---

You are an expert at creating modified queries with intentional irrelevance while maintaining the core context of the tools and arguments provided.

Each of the answer is expected to follow the format below:

Query1: (irrelevance added query)

Query2: (irrelevance added query)

Queryn: ....

You will receive some queries along with a list of tools, including their descriptions and required parameters. You will also be provided with an example of original vs augmented query. Your task is to introduce irrelevance by creating queries that either:

1. Include a constraint or condition not relevant to any of the listed arguments, or

2. Request execution of one function but provide parameters meant for a different function.

3. Make a completely new request which is unrelated all the list of functions provided.

When modifying each query, follow these guidelines:

1. Ensure that the added irrelevance does not alter the core meaning but introduces a new element or mismatch that makes the query more challenging to interpret.

2. On the original query section, the response should not hallucinate; it should remain constrained to the parameters available, even if they don't perfectly match.

3. Innovate by intentionally introducing subtle conflicts in argument requirements or irrelevant requests.

4. Maintain the provided format for each modified query.

ONE-SHOT EXAMPLE:

**Original Query**: Find articles from NASA or SpaceNews related to Artemis missions, not including those from European Spaceflight, limited to 2 results published between January 1st, 2021, and December 31st, 2021.

**Augmented Query**: Retrieve two articles specifically with summary title containing Artemis missions, exclusively from NASA or SpaceNews, excluding any pieces from European Spaceflight. Ensure the publication dates fall within January 1, 2021, to December 31, 2021. Set an offset of 4 for the search results.

---

Table 8: Sample Benchmark Execution Prompt

Based on the previous context and API request history, generate an API request
or a response as an AI assistant.
We detail the available tools with their respective name, description, in-
put(parameters) of each action as follows:

```
[
  {
    "name": "get_joke_of_the_day_by_category",
    "description": "Get the joke of the day of a specific category",
    "parameters": {
      "category": {
        "type": "string",
        "description": "Category of the joke from the jokes categories API.",
        "required": true
      },
      "limit": {
        "type": "number",
        "description": "Number of jokes to output.",
        "required": false
      }}},
  ...
  {
    "name": "finish",
    "description": "Return an answer and finish the task.",
    "parameters": {
      "answer": {
        "type": ["string", "number", "array"],
        "description": "Finish task using the answer parameter.",
        "required": true
      }}}
]
```

ONE-SHOT EXAMPLE:

```
Goal: Please List all the categories of jokes available and
provide me with 2 jokes of the day.

Trajectory:
Action: get_categories_of_joke with Action Input: {}
Observation: ['food', 'politics', 'art', 'sports']

Action: get_joke_of_the_day with Action Input: {limit: 2}
Observation: [{'id' : 1, 'result': A man staggers into an
emergency room with two black eyes and a five iron wrapped
tightly around his throat. Naturally the doctor asks him what
happened. } ...]
```

Your output should follow the provided format which specifies the specific func-
tion to call along with its respective input. The example format is as follows.
**Action**: [your action] with **Action Input**: [your action input]
If no function call is required additionally, please conclude the response using
the finish function.

Table 9: Sample Trajectory with Feedback

```
Goal: Could you give us a list of sports and a list of markets,
      starting from page 3 to see the variety of options?

Trajectory:
  Interaction 1:
    Action: list_of_markets_for_pinnacle_odds
    Action Input: {"sport_id": 1, "page_num": 3}
    Feedback: ERROR | Invalid argument name page_num for function
    list_of_markets_for_pinnacle_odds, Valid arguments
    are ['sport_id', 'league_ids', 'event_type', 'event_ids']
  Interaction 2:
    Action: list_of_sports_for_pinnacle_odds
    Action Input: {\"sport_id\": 1, \"page_num\": "3"}
    Feedback: ERROR | Invalid argument type for
        page_num for function list_of_sports_for_pinnacle_odds
        Valid argument type is 'int'.
  Interaction 3:
    Action: list_of_sports_for_pinnacle_odds
    Action Input: {\"sport_id\": 1, \"page_num\": 3}
    Observation: [{'id': 1, 'p_id': 29, 'name': 'Soccer',
            'last': 1688136047, 'special_last': 1688136033,
            'last_call': 1688136047}, {'id': 2, 'p_id': 33,
            'name': 'Tennis', 'last': 1688136023,
            'special_last': 1688118591,
            'last_call': 1688136041}].....
```

