# OpenReview forum: "ToolScan: A Benchmark For Characterizing Errors In Tool-Use LLMs"
_ICLR.cc/2025/Workshop/BuildingTrust — BuildingTrust_

### Official Review · Reviewer_VsNY · 2025-02-22
**A useful benchmark**

**Rating:** 7
**Confidence:** 4

**Review:**

This paper presents a new benchmark for fine-grained evaluation of the LLM-with-tools framework.

Strengths: The paper is well-written and clear, the benchmark is useful, and analysing tool-calling performance by error type allows more informative evaluations. The authors test multiple, new and SoTA LLMs.

Weaknesses: The evaluated LLMs are all base models prompted in context. It would be interesting to compare these with fine-tuned tool-augmented LLMs (e.g. Toolformer, ToolkenGPT etc.) and examine how the error patterns may change.

---

### Official Review · Reviewer_V8aV · 2025-03-01

**Rating:** 8
**Confidence:** 4

**Review:**

This paper provides an important benchmark for studying tool-use. Rather than just report correctness rates or progress rates in multi-step tool-use, this paper introduces a well-designed benchmark with very fine-grained metrics for evaluating tool-use. One con might be that it is automatically constructed, but I think automatic construction is reasonable for a tool-use benchmark and is not automatically a con.

---

### Official Review · Reviewer_SPqX · 2025-03-02
**ToolScan Review**

**Rating:** 6
**Confidence:** 3

**Review:**

## Contributions
The authors introduce a benchmark that augments existing agentic-flow API-call benchmarks. They formalize the common reasons that agents fail on these cases, and present a framework that categorizes the trajectories into the respective failure modes. They present results from ablation studies on many SOTA models.

## Strengths
- Augmented datasets with rephrasing + added confusion/complexity, since LLMs need to be able to function even when the instructions are not perfectly clear
- Useful to break down the failure cases into more granular failure reasons for interpretability
- Interesting ablation study results reported
## Weaknesses
- Some minor typos (weather and movie)
- The entire dataset, across so many fields, only has 150 cases?
- The mathematical description of LLM agents is likely not necessary, since many papers in this space have already formalized their mathematical underpinnings
- Does not include any Anthropic models

## Questions
- How are you determining which of the seven failure cases each trajectory reaches? A longer description of the methodology of that component of your framework would be helpful.

---

### Decision · Program_Chairs · 2025-03-04

Accept